# DNA Methylation in Alcohol Use Disorder

**DOI:** 10.3390/ijms241210130

**Published:** 2023-06-14

**Authors:** Qingmeng Zheng, Heng Wang, An Yan, Fangyuan Yin, Xiaomeng Qiao

**Affiliations:** 1Department of Pathology and Forensic Medicine, School of Basic Medical Sciences, Zhengzhou University, Zhengzhou 450001, China; zqm19961027@126.com (Q.Z.);; 2School of Medicine, College of Forensic Science, Xi’an Jiaotong University, Xi’an 710061, China

**Keywords:** alcohol, DNA methylation, gene transcription, age, inhibitors

## Abstract

Excessive drinking damages the central nervous system of individuals and can even cause alcohol use disorder (AUD). AUD is regulated by both genetic and environmental factors. Genes determine susceptibility to alcohol, and the dysregulation of epigenome drives the abnormal transcription program and promotes the occurrence and development of AUD. DNA methylation is one of the earliest and most widely studied epigenetic mechanisms that can be inherited stably. In ontogeny, DNA methylation pattern is a dynamic process, showing differences and characteristics at different stages. DNA dysmethylation is prevalent in human cancer and alcohol-related psychiatric disorders, resulting in local hypermethylation and transcriptional silencing of related genes. Here, we summarize recent findings on the roles and regulatory mechanisms of DNA methylation, the development of methyltransferase inhibitors, methylation alteration during alcohol exposure at different stages of life, and possible therapeutic options for targeting methylation in human and animal studies.

## 1. Introduction

Alcohol abuse has become a serious public health and biomedical problem. It is responsible for more than 3 million deaths worldwide each year (5.3 percent of all deaths) and 5.1 percent of burden of disease, with a financial cost of $250 billion [1]. Long-term abuse of alcohol induces alcohol use disorders (AUD), accompanied by a large number of molecular and biochemical changes [2]. As a chronic relapsing brain disease, AUD is associated with genetic and environmental factors [3,4]. Susceptibility to alcohol is determined by genes, while epigenetic mechanisms regulate chromatin structure by integrating different environmental stimuli, resulting in strong and lasting changes in gene expression, thus controlling the development of AUD [5,6]. Emerging evidence suggests that epigenetic modifications affect gene transcription and expression in cell cycle, signal transduction and other processes, then changes the physiological and pathological processes of the brain [7,8]. In particular, methylation of the cytosine site in CpGs is among the best-characterized epigenetics that mediates spatiotemporal specific changes in gene promoters, which are potentially reversible and can be passed on to offspring through multiple cell divisions [9,10]. The strong heritability of alcohol abuse suggests the existence of heritable changes in the function of genes that alter alcohol metabolism or neuronal plasticity and the neurobiology of reward, cognitive, and anxiety/depression. Therefore, the study of DNA methylation in alcohol abuse is helpful to clarify the genetic and molecular mechanisms and provide new insights for clinical treatment of AUD. This review summarizes recent findings on DNA methylation alterations underlying development of alcohol abuse/AUD.

## 2. DNA Methylation and Its Regulatory Mechanisms

DNA methylation is an important epigenetic marker involved in the life process of *eukaryotes* [11]. In *vertebrates*, DNA methylation underlies regulatory mechanisms such as *embryonic* development and *cell* reprogramming [12]. DNA methyltransferases (DNMTs) transfer a methyl group of S adenosylmethionine (SAMe) to the fifth carbon atom of cytosine to form 5-methylcytosine (5mC), participating in long-term silencing of genes [13]. More than 80% of CpG sites in the *human* genome are scattered and highly methylated [14]. The 0.5–5 kb DNA fragments with CpG dinucleotide clusters in the 60–70% GC-rich DNA region are designated as CpG islands, which are located in the first exon and promoter of the gene [15]. CpG islands are usually unmethylated and highly conserved, and about 70% of promoters contain CpG islands. CpG island methylation in promoter region regulates gene transcription through multiple mechanisms [16]. For example, the transcriptional activity of genes is suppressed by blocking the binding of *transcription factors (TFs)* to the promoter region, or inhibition of transcription by Methylated CpG site-binding proteins to recruit co-inhibitory complexes [17].

### 2.1. Methylation and Demethylation

DNA methylation is catalyzed and maintained by DNMTs, which include two categories: (1) De novo methylases DNMT3A and DNMT3B are enzymes that establish initial methylation patterns on unmethylated DNA. DNMT3 plays an important role in early *embryonic* development and normal cellular differentiation [18]. (2) Maintenance methylase DNMT1 replicates methylation patterns from parent DNA strands to progeny strands during DNA replication [19]. DNMT1 knockout in *mice* leads to DNA methylation loss, cell apoptosis, and *embryonic* death [20]. Another specific DNMT enzyme is DNMT3L, which is expressed only in *adult* germ *cells* and early developing thymus. DNMT3L is a non-catalytic protein that combines DNMT3A and DNMT3B with methyltransferase activity. In *mice*, DNMT3L participates in establishing genomic imprinting, reverse methylation transfer, and X chromosome agglutination between offspring and parents [21,22].

Methylation of CpG sites in the promoter can be specifically recognized by some methylated CpG binding domains (MBDs), which mainly include MBD1-4 and methylated CpG binding protein 2 (MeCP2) [23]. MBD contains transcriptional inhibition domains (TRD) that bind to various repressor complexes to inhibit transcription [24]. MeCP2 can recruit DNMT1 to semi-methylated DNA to maintain methylation [25]. MeCP2 also binds to CpG sites and recruits transcription inhibitor complexes, such as DNMTs and histone deacetylases (HDACs), thus inhibiting gene transcription [23].

There are two ways of DNA demethylation: passive and active demethylation. During *cell* division, *cells* can block DNA maintenance methylation by inhibiting DNMT1 expression or catalytic activity, and achieve passive demethylation by diluting/reducing the density of methylated cytosine in the genome [26]. The majority of DNA is passively demethylated during cleavage. Active DNA demethylation is mainly dependent on the TET family (ten-eleven translocation enzymes, TET1/2/3) and thymine-DNA glycosylase (TDG). The TET enzyme oxidizes 5mC and 5-hydroxymethylcytosine (5hmC) to 5-formylcytosine (5fC) and 5-carboxylcytosine (5caC), while TDG is responsible for the selective identification and removal of 5fC and 5caC, restoring them to common cytosine by base excision repair. 

### 2.2. Mechanism of DNA Methylation Regulating Gene Transcription

DNA methylation regulates gene transcription mainly through three mechanisms (Figure 1). First, 5-methylcytosine (5mC) is located in the major groove of DNA, thus occupying the site where transcription factors bind to DNA and hindering gene transcription [27]. Second, CpG sites in DNA promoter region can be specifically identified by some MBDs after methylation. After binding to CpG site, MBDs recruit DNMTs, histone methyltransferases (HMTs) or histone deacetylases (HDACs) to form co-inhibitory complexes, thereby inhibiting gene transcription [28,29]. Third, chemical modifications like acetylation, phosphorylation and ubiquitination occur at specific amino acid residues at the N-terminal of histones coated with methylated DNA, which affect recruitment of DNA-binding proteins and chromatin structure, resulting in looser or tighter chromatin around specific loci [30]. In addition, methylated cytosine may inhibit gene expression by reducing RNA polymerase activity through some mechanism [31]. Recent studies also found a mutual regulation between DNA methylation and miRNA. Methylation of CpG island in the promoter regions of miRNAs can inhibit their transcription [32]. Conversely, miRNA regulates methylation by affecting DNMTs directly [33]. In summary, it is believed that DNA methylation, histone modification and miRNA may co-regulate gene expression and biological processes through complex interaction, and this interaction can be significantly affected by alcohol.

### 2.3. DNA Methylation Inhibitors

Dysregulation of DNA methylation is prevalent in cancer and psychiatric disorders, leading to local hypermethylation of the gene concerned and subsequent transcriptional silencing. Epigenetic reversibility allows it to be a target for drug design. DNA methylation inhibitors include the following categories: Cytidine analogs, DNA binders, oligonucleotides and polyphenols (Table 1). Cytidine analogs include: 5-aza [34], RX-3117 [35] or other synthetic nucleoside analogues. They incorporate DNA during replication, competitively preempt DNMTs with cytosine, and covalently bind to sulfhydryl groups on cysteine residues of DNMTs, thus inactivating them [36]. Over the years, more and more stable cytidine analogs with low toxicity and strong specificity have been designed, such as 5-aza-dc [37], zebularine [38]. Several small molecule inhibitors such as RG-108 [39], SGI-1027 [40] and GSK3685032 [41] bind non-covalently to DNMTs active sites, inducing demethylation, transcriptional activation, and cancer cell growth inhibition. Other inhibitors can bind to CpG sites and DNMTs, such as procainamide [42] and procaine [43]. Oligonucleotides (MG98 [44] and miR29b [45]) consist of 15–30 nucleotides, which complement the additional coding RNA of some genes and bind to the SAM cofactor of DNMTs, inhibiting DNMTs expression [46]. In addition, polyphenols act by non-covalently binding to the active site of DNMT enzyme, Such as EGCG [47], curcumin [48], γ-oryzanol [49]. DNA methylation inhibitors have strong development and application prospects in the treatment of cancer and mental diseases, and need further clinical research.

## 3. DNA Methylation and Alcohol Abuse

Genetic factors play an important role in the development of AUD. Family and twin studies have shown AUD to be 50–65% heritable [50]. The complex etiology of AUD lies not only in the variation of candidate genes, but also may be caused by the change in epigenetic modification. Epigenetic modification controls the occurrence and phenotype of diseases, and can be passed on to offspring [51]. Epigenetic mechanisms rearrange under environmental stimuli, resulting in efficient and lasting changes in gene expression, thus promoting alcohol-induced transcription and behavioral changes [52]. In *mammalian* neurogenesis and brain formation, DNA methylation plays a role in determining the timing and extent of gene transcription [53]. Changes in DNA methylation profiles are closely associated with neurodysplasia, cognitive and behavioral impairments, and psychiatric disorders [54]. Therefore, this review focuses on the role of DNA methylation in *prenatal*, *adolescent*, and *adult* alcohol exposure.

### 3.1. DNA Methylation Changes Ethanol Oxidation System

Most alcohol is metabolized by alcohol dehydrogenase (ADH) catalyzed oxidation system and microsomal ethanol oxidizing system (MEOS) in liver [55]. The ADH oxidation system metabolizes alcohol into acetaldehyde and acetic acid, and MEOS will be active if ADH system cannot quickly remove ethanol by long-term heavy drinking. The core element of MEOS is cytochrome P450 family 2 subfamily E member 1 (CYP2E1), which requires oxygen for its catalytic process, so it produces a series of free radicals that damage liver. Excessive alcohol consumption increases the expression and activity of CYP2E1 to activate carcinogens and hepatotoxins, converting them into more toxic metabolites [56]. DNA methylation of CYP2E1 plays an irreplaceable role in maintaining normal liver function. Southern blot analysis revealed significant methylation of the cytosine 3′ region of the CYP2E1 gene in *adult* and *fetal* liver samples [57]. CYP2E1 specific probe clozazoldone clearance was detected in liver microsomes, and it was found that CYP2E1 intrinsic clearance was correlated with DNA methylation and H3K9ac levels [58]. Reactive oxygen species (ROS) produced by CYP2E1 binds to DNA to form complexes through lipid peroxidation, both of which can regulate the methylation level of DNA [59]. Moreover, alcohol reduces the methylation level of CYP2E1, resulting in decreased cell activity and death through ROS pathway in mouse *embryonic cells* [60]. In alcoholic hepatitis, alcohol promotes CYP2E1 methylation via the methionine adenosyltransferase α1 (MATα1) pathway, a methyl donor responsible for liver biosynthesis of S-adenosylmethionine. MATα1 interacts with CYP2E1, and the methylation level of CYP2E1 increases at the R379 site, which affects cell proliferation and apoptosis [61]. Folic acid plays an important role in DNA methylation modification, and its metabolite S-adenosylmethionine (SAM) is a methyl donor. Studies have shown that folic acid regulates the expression and catalytic activity of CYP2E1 and acetaldehyde dehydrogenase (ALDH, catalyze ethanol into acetaldehyde). Under the action of folic acid, harmless substances of alcohol metabolism are excreted from the body, resulting in reduced DNA damage and changed methylation levels [62]. In addition, CYP2E1 is involved in the occurrence and development of neurological diseases. It was found that the low methylation level of CYP2E1 may be related to the underlying mechanism of schizophrenia [63]. Evaluation of candidate genes by nitrite phosphosequencing in cortical tissue of patients with Parkinson’s disease revealed hypomethylation of CYP2E1 [64]. In conclusion, CYP2E1 induces DNA damage and methylation changes, and its own methylation is involved in the regulation of alcohol-induced brain and liver disorders, which may be a potential therapeutic target.

### 3.2. DNA Methylation Profiles in Prenatal Alcohol Exposure (PAE)

Early pregnancy is a dynamic period of epigenetic reprogramming, cell divisions, and DNA replication and, therefore, *prenatal* alcohol abuse increases the susceptibility of spontaneous miscarriage, sudden *infant* death and *fetal* alcohol spectrum disorders (FASD) in offspring [65,66]. DNA methylation is a key factor in epigenetic regulation of *mammalian* embryonic development. Before fertilization, mature sperm and oocytes remain highly methylated. The parental genome undergoes a second cycle of demethylation from fertilization to the mulberula stage before embryo implantation, followed by progressive re-methylation [67,68]. Furthermore, the inverse correlation between the degree of DNA methylation and gene expression reaches its peak at the post-implantation stage [69]. *Human* and *animal* studies have suggested that DNA methylation is a potential mediator and biomarker for the effects of PAE because of its response to environmental cues and relative stability over time, though the molecular mechanisms involved are poorly understood.

Alcohol profoundly and extensively alters DNA methylation patterns in a developing *embryo*. Shayan Amiri found that alcohol (70 mmol/L alcohol solution for 8 days) induced global DNA hypomethylation in Neural Stem Cells (NSC), while DNMTs and TETs expression were altered in a sex- and strain-specific manner, ultimately leading to differential and developmental disability of NSC [70]. An earlier study showed that alcohol induced hypermethylation of genes on chromosomes 7, 10 and X during early neurulation in a whole *embryo* culture; in contrast, the methylation rate of genes with high CpG promoters was lower, but the expression rate was greater [65]. PAE-induced changes in DNA methylation during embryonic development may be the cause of FASD, as these genes are enriched with neurodevelopmental functions. The hypothalamus and leukocyte DNA methylation profiles of the offspring of PAE female rats were persistently altered, with some genes overlapping with the developmental profile findings [71]. On temporal lobe samples of autopsied *fetuses/infants* and 5.7 to 6 months old *macaque monkeys* with documented PAE, 5mC, H3K4me3, 5fC and H3K36me3 were significantly decreased in the DG and the ependyma, leading to neurodevelopmental abnormalities and *infant* death [72]. PAE can reduce the volume and alter the electrophysiological characteristics of brain, resulting in learning, memory defects and behavioral changes. *Solute carrier family 17 member 6 (Slc17a6)*, which encodes vesicular glutamate transporter 2 (VGLUT2), was upregulated in hippocampus of male offspring of PAE *mice*, and this upregulation was associated with reduced DNA methylation H3k4me3 enrichment [73]. Moreover, it was found that PAE can lead to changes in the overall DNA methylation of the prefrontal cortex (PFC) and hippocampus [74]. DNA methylation program proceeds along with the differentiation and maturation of hippocampal neurons. In *fetal* hippocampal CA1 neurons, PAE blocked the acquisition and progression of 5mC and 5hmC, which was related to developmental delay [75].

Emerging evidence suggests DNA methylation induced by PAE causes intellectual impairment, adaptive dysfunction, learning and memory deficits in the later stage of the *fetus*. However, the molecular pathways behind these long-lasting effects still need to be studied. One possible mechanism is that alcohol reduces maternal levels of folate and vitamins B6/12, which are involved in homocysteine metabolism. The absence of methyl donors alters the establishment of epigenetic markers in developing embryos [76,77]. Epigenetic changes in the first *embryonic cells* can be fixed in a lasting cellular memory and transmitted mitotically to different *cell* and tissue types [78]. Therefore, changes in DNA methylation during *embryonic* development will contribute to the complex phenotype of FASD. Alcohol also induces hypermethylation of multiple *cell* cycle genes related to G1/s and gap 2/mitotic phase (G2/M), and increases the expression and activity of DNMTs. Finally, alcohol affects *cell* cycle progression and nerve development [79,80]. Brain neurodevelopment is highly dependent on the epigenome [81]. One study tested ethyl glucuronic acid (EtG) in the meconium of 156 primary school children who had abused alcohol in their mothers after birth, and found that 193 genes involved in neurodegeneration, neurodevelopment, axon guidance and neuron excitability genes were hypermethylated. As a result, these students had cognitive and attention deficits [82]. Furthermore, PAE can alter the methylation of imprinted genes. Genomic imprinting enables parent-of-origin specific monoallelic expression of a select set of genes that are important in early development, particularly neurodevelopment [83]. In alcohol-exposed 9.5 embryonic-day-old (E9.5) embryos and placentas, the methylation changes in imprinted genes Insulin-like *growth factor 2 (Igf2)*, *H19,* and *Paternally expressed gene 3 (Peg3)* led to growth restriction in the *embryo* [84]. Ultimately, it appears that DNA methylation landscape is one of the prime mechanisms brain gene expression following PAE.

### 3.3. DNA Methylation Changes during Adolescent Alcohol Exposure

*Adolescence* is a special period in which the body structure develops rapidly and the psychological development is relatively slow [85]. *Adolescents* have less self control, are more easily influenced by environment and others to abuse alcohol, tobacco and drugs [86]. The vast majority of *adolescents* first attempt to drink alcohol during *adolescence*, with the highest incidence of AUD later occurring between the ages of 12 and 14, this means adolescence is a “risk window” for first-time drinking [87]. *Adolescent* alcoholism is linked to a range of morbidity in later life, in which process changes in DNA methylation remain relatively stable over time, causing lasting damage.

Alcohol abuse in *adolescents* increases the risk of neuropsychiatric disorders, including alcoholism in *adulthood* [88]. DNA methylation responds sensitively to alcohol stimuli, interfering with neurogenesis and synaptic formation, which continues into adulthood [89]. After intermittent alcohol exposure during adolescence, the methylation levels of brain-derived *neurotrophic factor (Bdnf)* exon IV and *neuropeptide Y (Npy)* increased in the amygdala of *adult rats*, this was accompanied by high alcohol intake and anxiety-like behavior [90]. Increased site-specific CpG methylation of the *serotonin transporter (SLC6A4)* gene, which is sensitive to depressive and addictive behaviors and may lead to cognitive and behavioral abnormalities, has been found in the saliva of some alcohol-prone *adolescents* [91]. After *adolescent* intermittent ethanol exposure (AIE), *miR-137* increased and its target genes lysine-specific demethylase 1 (Lsd1 and Lsd1 +8a) decreased in the *adult* amygdala of *rats*. While *miR-137* antagomir rescued AIE-induced alcohol drinking and anxiety-like behaviors via normalization of decreased *Bdnf* IV and *Lsd1* expression through increasing H3K9 dimethylation in *adult rats* [92]. AIE also significantly reduced basal forebrain cholinergic (TrkA^+^, ChAT^+^) neurons that persist into adulthood, which is due to a persistent increase in *adult* DNA methylation of *TrkA* and *ChAT* promoter regions and H3K9me2, resulting in impaired spatial memory in rats [93]. When parents were exposed to alcohol during adolescence, their male *PND7* alcohol-naïve offspring exhibited differential DNA methylation patterns in the hypothalamus, and the methylated difference also depended on which parent was exposed to alcohol [94].

Because the *adolescent* brain and epigenetic mechanisms are highly sensitive to environmental stimuli, and DNA methylation can remain stable under certain conditions, the effects of alcohol are often observed in adulthood [20,95]. Furthermore, DNA methylation is not static. In the *adult* life of *rodents*, DNMT inhibitor 5-aza-dc could reverse hypermethylation at *Bdnf/Npy* and AIE-induced behavioral changes [90]. It is worth noting that epigenetic regulation is not simply a direct predictor in gene expression, and it is just a key driving factor in determining levels of gene expression [96]. The effect of AIE on gene transcription is very complex, involving epigenetic mechanism, signal transduction pathway and gene polymorphism, etc. Further studies are needed to tease out the complex interplay between alcohol, transcriptional direction, and epigenetic regulation.

### 3.4. DNA Methylation Changes Induced by Alcohol Abuse in Adulthood

Although most people try alcohol for the first time during adolescence [97,98], alcoholism is more common among the middle-aged and elderly [99,100]. *Adolescent* drinking stems from curiosity and impulsiveness, and is characterized by heavy drinking in a short period of time [101]. *Adult* drinking is a very complex psychosocial behavior, which is greatly influenced by family, environment and mentality [102]. As a result, *adults* with chronic alcoholism are more likely to develop alcohol dependence or alcohol use disorders (AUDs). Moreover, alcohol-induced negative affective state and cognitive decline are positively associated with age [103]. DNA methylation is spatiotemporally specific. At any stage of life, environmental or chemical stimuli can be integrated by this modification to make long-lasting change in gene expression by regulating chromatin structure [20]. Additionally, DNA methylation causes phenotypic changes by modifying the genetic structure of *adult* alcoholics.

Extensive changes in DNA methylation across the genome may vary with alcohol intake in adulthood. One 5606 Melbourne Collaborative Cohort Study showed that 1414 CpGs in blood were associated with alcohol intake. After 11 years, 513 of these CpG sites showed changes in methylation that were associated longitudinally with alcohol intake [104]. Another study identified 5254 differentially methylated CpGs in the PFC of 25 AUD individuals, in which the methylation of the *NR3C1* exon variant 1H encoding the glucocorticoid receptor was significantly increased, which might be the pathophysiological basis of AUD [105]. In a study that included 16 controls and 16 AUD postmortem human PFC subjects, 106 differentially methylated CpGs were mapped to 93 differentially expressed genes, including AUD related genes such as *GABRA1, GRIK3*, and *GRIN2C* [106]. Xu et al. identified 64 novel methylation sites associated with alcohol consumption in saliva *cells* from 1135 European American men [107]. In the peripheral blood genome of AUD patients, methylation of the 3′-protein-phosphatase-1G (PPM1G) promoter region increased, with decreased mRNA expression. The study also found that PPM1G was associated with increased impulsive behavior in 499 alcoholics [108]. Dopamine plays an important role in the reward mechanism of the cortical limbic circuit and is related to alcohol craving [109]. It has been found that the hypermethylation of dopamine transporter (DAT) gene promoter in the blood of alcoholics is negatively correlated with the desire for alcoholism [110]. These gene loci can be used as biomarkers of alcohol abuse.

Changes in alcohol-induced methylation and related regulatory factors have also been observed in many *animal* experiments. After three weeks of abstention, the DNMT1 levels of mPFC in alcohol-dependent rats continued to increase, while *synaptotagmin 2 (Syt2)* gene expression was downregulated. However, DNA methyltransferase inhibitor RG108 restored *Syt2* expression by inhibiting hypermethylation on CpG#5 of its first exon [111]. Similarly, Cui et al. found that the overall methylation level of mPFC in chronic alcohol exposure rats was significantly higher than that in the control group, accompanied by increased DNMT3B and MeCP2 levels. At the same time, the possible target genes such as *Ntf3*, PPM1G and *Dual Specificity Phosphatase 1 (DUSP1)* were screened [112]. Another recent study also demonstrated elevated global DNA methylation and hydroxymethylation levels in NAc and increased DNMTs activity in alcohol-preferring *rats* [113]. Studies have shown that alcohol impairs methionine synthase (Ms) activity, resulting in a reduced S-adenosyl methionine/S-adenosyl homocysteine (SAM/SAH) ratio and DNA hypomethylation [114]. In the cerebellum of long alcohol-exposed rats, SAM levels, SAM/SAH ratio, Ms and methylene tetrahydrofolate reductase were all decreased, which caused a series of changes in carbon metabolism, increased “methylation index” in cerebellum, and led to decreased expression of synaptic plasticity related genes and behavioral changes. However, one-carbon metabolism returned to near-normal levels during alcohol withdrawal [115]. Thus, it can be seen that DNA selectively responds to environmental factors and changes transcriptional mechanisms, and participates in various life processes such as cell cycle regulation and signal transduction.

Precise regulation of DNA methylation is essential for normal cognitive function. Long-term drinking in *adults* will change the normal structure and function of the central nervous system, resulting in memory decline, coding disorders, decreased flexibility, impulsive behavior, anxiety, depression and so on. These neuropsychiatric abnormalities are closely related to the regulation of DNA methylation. *C57BL/6J mice* were exposed alcohol for 3 weeks, their fear memory and recognition memory were impaired, while methylation of the BDNF promoter region in the hippocampal CA1 region was reduced, and the *BDNF* signaling pathway mediated by ERK, Akt and CREB was upregulated to counteract alcohol-induced cognitive deficits [116]. The c-Jun NH(2)-terminal kinase (JNK2) was activated in the PFC of binge alcohol withdrawal (BAW) *mice*, and activation of JNK2 causally enhanced total genomic DNA methylation via increased DNMT1 expression. In addition, 5-aza-dc or JNK2-specific inhibition was shown to completely abolish BAW-evoked anxiety-like behavior [117]. Patients with AUD present with important emotional impairments such as depression [118]. Methylation of *NR3C1 glucocorticoid receptor (GR)* gene is associated with depression, post-traumatic stress and anxiety [119]. Therefore, it is reasonable to believe that *NR3C1* methylation regulation is a potential biological marker of AUD-induced depression.

## 4. DNA Methylation as a Therapeutic Target for AUD

Alcohol-induced epigenetic modification is a promising field for studying changes in specific gene promoter sites and genome-wide DNA methylation patterns. Numerous studies have demonstrated that DNA methylation can be used as a marker for cancer diagnosis and a target for disease therapy. Factors affecting methylation include DNA itself, methyl donor S-adenosyl-L-methionine (AdoMet), enzymes and cofactors [120]. Therefore, drugs are selected based on the synthetic material of DNA methylation to provide therapeutic targets (Table 2). Acetaldehyde, a metabolite of alcohol, causes DNA point mutations, double-strand breaks, sister chromatid exchange and chromosome structural changes, hindering DNA synthesis and repair and changing methylation level. Moreover, acetaldehyde inhibits the activity of DNMTs [121]. Acetaldehyde dehydrogenase (ALDH) alleviates DNA damage and promotes DNA repair, which is expected to regulate DNA methylation levels [122]. Therefore, inhibition of acetaldehyde may ameliorate alcohol-induced behavioral damage to some extent by affecting DNA methylation. In addition, several molecular proteins are used to regulate DNA methylation levels. Fanconi anemia gomplementation group D2 (FANCD2) protein slows DNA damage, maintains cell activity, and may modulate AUD [123,124]. Glutathione (GSH), the most abundant non-protein mercaptan, is involved in cell methylation metabolism through the sulfur transfer pathway and is used to treat AUD [125].Folates regulates methylation by providing a single carbon donor for methionine and nucleotide synthesis. Deficiencies in the diet of choline, methionine, vitamin B12 and folate lead to methyl-deficiency [126]. Folates supplementation relieves alcohol induced Th17/Treg disbalance through altering *Forkhead box O3(Foxp3)* promoter methylation patterns, and this effect may be caused by decreased DNMT3a [127]. Folates are not only essential for adults, but also control the survival and development of embryos, participate in single-carbon metabolism and transfer of one-carbon units, promotes DNA synthesis and DNA methylation cycle [128,129]. Oral folates in pregnant women may reduce nerve cell apoptosis caused by prenatal alcohol intake and prevent fetal alcohol syndrome [130]. Folic acid and vitamin B6/B12 are widely used to treat alcohol dependence, cognitive decline, and alcohol-induced liver damage [127,131]. These studies suggest that folates may be a viable preventive strategy for AUD.

Moreover, scientists have found that traditional Chinese medicine (tcm) also shows great potential in the treatment of alcoholic diseases by regulating DNA methylation. Curcumin is a plant extract that regulates the lifespan of alcohol-fed bees by increasing overall DNA methylation levels [132]. Betaine provides methyl to homocysteine (Hcy) to synthesize methionine, thereby correcting abnormalities in the methionine cycle and sulfylation to reduce alcohol-induced liver damage [133]. Ganoderma lucidum and cordyceps alcohol extract inhibit apoptosis and protect the liver by modulating methylation, which may be useful in the treatment of Alcoholic hepatitis and neuroinflammation [134,135,136]. Therefore, some factors in plant extracts, which are related to the methylation modification induced by alcohol and have no cytotoxicity, should be paid more attention in the treatment of alcohol-related diseases.

Though methyltransferase inhibitors were first developed for their anticancer effects [137]. scientists are also discovering that DNMTs is closely related to memory regulation, anxiety-like behavior, and alcohol-seeking behavior. RG108 prevented compulsive drinking behavior in *rats* by reversing hypermethylation on CpG#5 of *Syt2* first exon [111]. Our previous study also found that 5-aza-dc injection into the mPFC significantly decreased alcohol consumption and preference in *rats* [138]. Yang et al. confirmed that knockout JINK2 improved anxiety-like behavior and impared contextual associative memory induced by binge alcohol withdrawal through offsetting c-JUN-regulated DNMT1 upregulation and restoring DNA methylation levels in mouse PFC to baseline levels [117]. In alcoholic liver disease (ALD) progression, alcohol promotes hepatocyte apoptosis and DNA damage by reducing TET1-mediated 5hmC formation and DNA methylation [139]. Iron is a cofactor of TET enzymes that catalyze the conversion from methylcytosine to hydroxymethylcytosine. Adding carbonyl iron to the diet of chronic alcohol-exposed *rats* reversed low DNA hydroxymethylation levels in the liver [140]. In short, scientists have been using various methods to normalize alcohol-induced DNA methylation in animals, with the aim of treating alcohol-related diseases.

DNA methylation may be an ideal target for drug therapy, but related drugs still have obvious disadvantages. First, the targeting ability of DNMTs inhibitors is not strong, they tend to change the overall methylation level and lack gene specificity, which greatly reduces the accuracy of drugs. There is no significant correlation between methylation/demethylation and clinical efficacy [141]. Second, the process of DNA methylation is reversible, and once demethylated drugs are stopped, the disease may relapse, so the drug must be continued after the benefit is obtained. Moreover, the inherent cytotoxic effects of DNMTs inhibitors should not be ignored, which limits the clinical application. Therefore, the combination of different modified inhibitors for the treatment of alcohol-related diseases is also gradually developed. Studies have shown that the combination of FDA-approved 5-aza-dc and HDAC inhibitor SAHA can effectively inhibit the motivation of alcohol seeking in rats and mice without damaging their metabolism [142]. Another study revealed that neonatal administration of thyroxine and metformin in patients with FASD improved memory impairment via elevating DNMT1 and consequently normalizing hippocampal deiodinase-III (Dio3) and insulin-like *growth factor 2 (Igf2)* expressions in the adult offspring [143]. Additionally, alcohol exposure during the fetal period increases the susceptibility to tumor, possibly by enhancing the methylation of *dopamine D2 receptor (D2R)* gene promoter and repressing the synthesis and control of *D2R* on prolactin-producing cells. When fetal alcohol exposed rats were treated neonatally with 5-aza-dc and HDAC inhibitor trichostatin-A their pituitary *D2R* mRNA, pituitary weights and plasma prolactin levels were normalized [144]. Table 2 summarizes the drugs that may target DNA methylation for AUD.

Demethylation drugs have been widely used in clinical practice, but there are still many misunderstandings regarding rational drug use. It is very important to identify the gene sites of methylation before treatment, except for choosing the appropriate dosage and course of treatment. If a specific gene regulated by methylation is found, it may be more effective to use advanced gene editing or transgenic techniques to interfere with the gene itself. Both pharmacological and behavioral therapies of AUD are underutilized, and therefore, given that many treatment options are still not incorporated into evidence-based practice, more research on dissemination and implementation is urgently needed [145].

## 5. Conclusions

The work reviewed here provides evidence that DNA methylation may play an important role in AUD by regulating gene transcription. AUD influences behavior through genetic and environmental factors, seems to be an interesting avenue to study. The relative stability and heritability of DNA methylation are responsible for the influence of parental alcoholism on offspring behavior and cognition, and the negative effect of *adolescent* alcohol exposure on *adult*. Studying the effects of alcohol abuse on methylation levels in specific gene promoterat different stages and gene expression manipulation could lead to a deeper understanding of epigenetic mechanisms. The development and promotion of DNA methylation inhibitors provide feasible ideas for the prevention and treatment of AUD, thus paving the way for new fields of investigation and treatment. It is worth noting that the development of AUD depends on multiple aspects and calls for the contribution of more disciplines. In the future, more advanced molecular biology techniques and multidisciplinary cooperation are needed to study the mechanism and treatment of AUD.

## Figures and Tables

**Figure 1 ijms-24-10130-f001:**
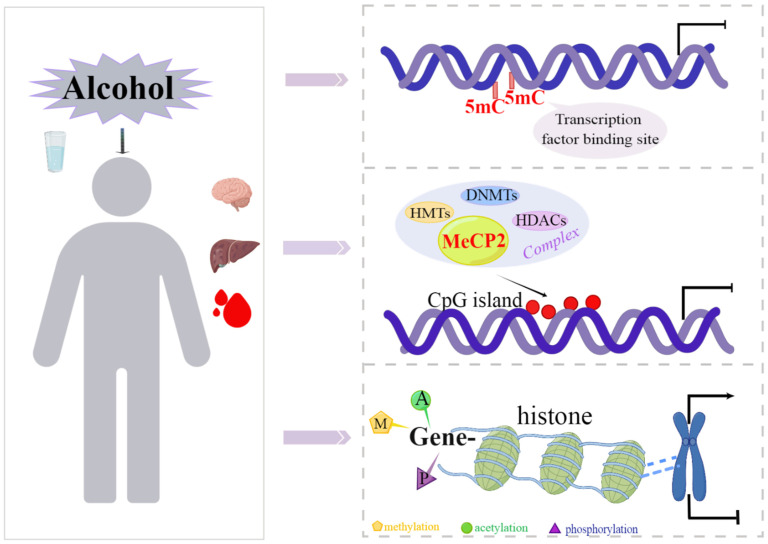
Alcohol-induced changes in DNA methylation and its transcriptional regulation mechanism.

**Table 1 ijms-24-10130-t001:** Representative DNA methyltransferase inhibitors.

Classification	Drugs	Mechanisms	Status of Clinical Use	Refs.
Cytosine nucleoside derivatives	5-aza RX-3117	Incorporate DNA and participate in DNA replication	clinical application preclinical study	[34,35]
Deoxyribose analogs	5-aza-dc	incorporate DNA and participate in DNA replication	clinical application	[37]
Benzoamide	Zebularine RG108 SGI-1027	bind non-covalently to the active sites of DNMTs	preclinical study preclinical study preclinical study	[38,39,40]
Aminobenzoic acid derivatives	Procainamide Procaine	bind to the CpG sites	Clinical phase II Clinical phase II	[42,43]
Antisense oligonucleotide Polyphenols	MG-98 miR29a EGCG curcumin γ-oryzanol	act on DNMT1 mRNA Bind the active site of DNMT enzyme, bind to DNMT sulfhydryl	Clinical phase II preclinical study preclinical study preclinical study preclinical study	[44,45,47,48,49]

**Table 2 ijms-24-10130-t002:** Representative drugs for AUD.

Drugs	Mechanism	Function	Status of Clinical Use	Refs
**ALDH** **FANCD2** **Glutathione** **Folic acid** **Vitamin B6/B12**	Reduce DNA damage and maintain DNA activity Reduce DNA damage and maintain DNA activity Neutralize free radicals, transport cysteine, and REDOX cells Methyl donor Methyl donor	Improve hematopoietic function, protect the liver Improve hematopoietic function, protect the liver Promote cell regeneration Improve oxidative damage and cognitive impairment Improve oxidative damage and cognitive impairment	preclinical study preclinical study clinical application clinical application clinical application	[122,123,124,125,127,131]
**Curcumin** **betaine** **Lucidum** **Cordyceps sinensis**	Increase methylation levels provide methyl groups to make S-adenosine Increase the expression of histone H3, DNMT3A and DNMT3B. Promote DNA methylation reprogramming	Resist cell oxidative damage and apoptosis Repairing damage to embryonic development Improve oxidative damage and cognitive impairment Improve brain atrophy and learning and memory function	clinical application clinical application clinical application clinical application	[132,133,134,135,136]

## Data Availability

No new data were created or analyzed in this study. Data sharing is not applicable to this article.

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
