# Peer review of "DNA Methylation in Alcohol Use Disorder"

_ijms, 2023, doi:10.3390/ijms241210130_

Round 1

Reviewer 1 Report

The authors conducted a serious analysis of the literature and made a very good review article. The main research question is DNA methylation during alcohol consumption. Its influence on the reading of information from DNA, as well as the analysis of the contribution of DNA methylation to the development of oncology and pathologies in offspring. The authors reviewed the latest research in this area in detail and combined them into a review. In addition, the mechanisms of the influence of DNA methylation on the occurrence of genetic problems have been proposed.

However, it has a few notes:

1. In the text, there are very often places with no spaces before reference to the literature, for example: “... maintain methylation[25].”

2. Line 164 says: "Shayan Amiri found that alcohol (70mM/8d) induced global DNA...". What does 70mM/8d mean? Needs to be decrypted.

3. In line 219, in the fragment “…first-time drinking[78, 79].xAdolescent…” there are no spaces and there is “x”.

Author Response

Comments and Suggestions for Authors

The authors conducted a serious analysis of the literature and made a very good review article. The main research question is DNA methylation during alcohol consumption. Its influence on the reading of information from DNA, as well as the analysis of the contribution of DNA methylation to the development of oncology and pathologies in offspring. The authors reviewed the latest research in this area in detail and combined them into a review. In addition, the mechanisms of the influence of DNA methylation on the occurrence of genetic problems have been proposed.

However, it has a few notes:

  1. In the text, there are very often places with no spaces before reference to the literature, for example: “... maintain methylation[25].”

Response 1: The full article was reviewed in detail, and all references are preceded by spaces.

  1. Line 164 says: "Shayan Amiri found that alcohol (70mM/8d) induced global DNA...". What does 70mM/8d mean? Needs to be decrypted.

Response 2: 70mM/8d refers to treatment with 70 mmol/L alcohol solution for 8 days continuously. We added this information to the appropriate location in the article (Line 281).

  1. In line 219, in the fragment “…first-time drinking[98, 99].xAdolescent…” there are no spaces and there is “x”.

Response 3: This error has been corrected (Line 384).

Reviewer 2 Report

The review "DNA methylation in alcohol use disorder" has significant importance in the studies of disorders caused by alcohol. This review has well cited and included necessary information of DNA methylation role alcohol related disorders. Hence i accept the review in its present form for publication.

The manuscript need to be checked by native english speaker has i found some grammatical errors. Also need to check spacings, puntuations, commas and other marks whether placed properly or not.

Author Response

Reviewer 2: Comments and Suggestions for Authors

The review "DNA methylation in alcohol use disorder" has significant importance in the studies of disorders caused by alcohol. This review has well cited and included necessary information of DNA methylation role alcohol related disorders. Hence i accept the review in its present form for publication.

Points: Comments on the Quality of English Language

The manuscript need to be checked by native english speaker has i found some grammatical errors. Also need to check spacings, puntuations, commas and other marks whether placed properly or not.

Response : Thanks very much for your comments. We have invited a native English-speaking colleague to polish the language and format of this review and revised the whole paper according to suggestions. Please see if the revised version met the English presentation standard.

Reviewer 3 Report

Dear authors, congratulations for your performed work in preparing this manuscript.

The article provides a concise overview of the subject's main focus, which is the relationship between DNA methylation and alcohol use disorder (AUD). It effectively highlights the importance of genetic and environmental factors in the development of AUD, as well as the role of DNA methylation in driving abnormal transcription programs.

It adequately describes DNA methylation as an early and extensively studied epigenetic mechanism, emphasizing its stability and inheritance. It also mentions the dynamic nature of DNA methylation patterns during ontogeny, which is crucial for understanding its implications in AUD.

Furthermore, the article highlights the prevalence of DNA dysmethylation in human cancers and alcohol-related psychiatric disorders, resulting in hypermethylation and transcriptional silencing of relevant genes. This context helps to establish the relevance of DNA methylation in the pathogenesis of AUD.

It also mentions the development of methyltransferase inhibitors and their potential therapeutic applications, suggesting that targeting methylation may hold promise for treating AUD. This discussion of potential therapeutic options adds value to the abstract, as it hints at the practical implications of the research.

Overall, the manuscript effectively outlines the main points of the theme and provides a clear and concise summary of the topic. It covers the key aspects of DNA methylation, its role in AUD, and potential therapeutic avenues, which should generate interest among readers of the International Journal of Molecular Sciences.

More emphasis on the therapeutic drugs that can be used should be made, maybe through a table with available options under investigation or already used in clinical practice.

Round 2

Reviewer 1 Report

The authors have improved the article. I think that it can be accepted for publication in the journal.